# The Association of Pain Medication Usage and Quantitative Sensory Testing Outcomes in Fibromyalgia Patients: A Secondary Data Analysis

**DOI:** 10.3390/neurosci6010015

**Published:** 2025-02-10

**Authors:** Luana Gola Alves, Kevin Pacheco-Barrios, Guilherme J. M. Lacerda, Felipe Fregni

**Affiliations:** 1Neuromodulation Center and Center for Clinical Research Learning, Spaulding Rehabilitation Hospital, Harvard Medical School, Boston, MA 02138, USA; lgolaalves@mgb.org (L.G.A.); guilherme.lacerda@hc.fm.usp.br (G.J.M.L.); 2Vicerrectorado de Investigación, Unidad de Investigación para la Generación y Síntesis de Evidencias en Salud, Universidad San Ignacio de Loyola, Lima 15024, Peru

**Keywords:** fibromyalgia, pain medications, quantitative sensory testing

## Abstract

Background: Fibromyalgia syndrome (FMS), a chronic pain syndrome affecting 0.2–6.6% of the general population, is known for its challenging diagnosis and treatment. The known dysregulation in the Endogenous Pain Modulatory System (EPMS) characteristic of the pathology contributes to enhanced pain sensitivity. Fibromyalgia patients, who are often overmedicated, may experience, in addition to the drug-related known adverse effects, effects on fibromyalgia sensory-related outcomes. Therefore, the focus of this analysis is to explore the bidirectional drug–sensory outcome interactions, indexed by the conditioned pain modulation (CPM), an important assessment element in regard to an EPMS’s efficacy. Methods: Baseline data from a randomized, double-blind, single-center (Boston-based tertiary hospital) clinical trial (NCT03371225) were analyzed. Participants aged 18–65 with an FMS diagnosis and resistance to common analgesics were included. Demographic, clinical, and sensory variables, including CPM, temporal summation, and Pain-60 outcomes, were collected alongside a pain medication diary. Multivariable regression models adjusted for confounders were applied to explore associations between medication classes and quantitative sensory outcomes. Results: Out of 101 recruited FMS patients, we categorized the use of the following medications: antidepressants with 50% use (*n* = 50), muscle relaxants with 26% use (*n* = 26), and gabapentin with 25% use (*n* = 25). The results showed that antidepressant use correlated with worsened CPM, Odds Ratio = 0.39 (95% CI = 0.17–0.91), while muscle relaxants were linked to increased TSPS, β coefficient = 0.72 (95% CI = 0.0021–1.4431). On the other hand, gabapentin use was associated with elevated Pain-60, OR = 2.68 (95% CI = 0.98–7.31). Interestingly, the use of low doses of opioids was not associated with altered sensory measures. Conclusion: This cross-sectional analysis suggests that common pain medications may affect quantitative sensory outcomes in FMS patients. We provided important insights into bidirectional drug–sensory outcome interactions and their influence on pain medicine.

## 1. Introduction

Fibromyalgia syndrome (FMS) is a chronic pain condition. The literature shows a prevalence of 0.2–6.6% in the general population and 2.2–6.6% in women [1]. This disease impacts patients in multiple domains—including their lifestyle, daily responsibilities, and quality of life [2,3]. Adding to that, patients with FMS tend to have enhanced healthcare utilization. This pathology, characterized by general musculoskeletal pain, sleep deprivation, fatigue, and, in some cases, even cognitive disturbances [2], usually affects women, at a proportion of 9:1 [4], and is often misdiagnosed and misunderstood.

FMS most likely is associated with an alteration in the Endogenous Pain Modulatory System (EPMS), which consists of a complex network within the central nervous system [5] that is responsible for regulating nociceptive signals by inhibitory and facilitatory actions. In FMS patients, the EPMS appears to malfunction. Studies show irregularities in neurotransmitters [6,7] such as serotonin, norepinephrine, and dopamine, which are crucial biological features in pain modulation that contribute to the altered pain perception experienced by those with fibromyalgia. [8] The dorsal reticular nucleus and the caudal ventrolateral medulla are two medulla components that have significance in the pain modulation system. The first acts as a pronociceptive center, while the second acts as an antinociceptive center. The relation between these regions and the spinal dorsal horn is facilitated through closed reciprocal loops. In the spino-dorsal reticular nucleus loop, the ascending branch encounters strong inhibition by spinal GABAergic neurons, neutralizing the amplifying effect centered on the dorsal reticular nucleus. The ascending branch in the spino-caudal ventrolateral medulla loop, is excited by substance P (SP), which is released from primary afferents. In acute pain situations, it is believed that this SP-mediated excitation instigates an intense descending inhibition. However, during chronic pain, the lateral reticular formation within the caudal ventrolateral medulla is changed, transforming the action of the caudal ventrolateral medulla from inhibitory to excitatory upon meeting SP-responsive spinal neurons. Studies show a possible connection to a decreased expression of μ-opioid, δ-opioid, and GABA_β_ receptors.

Quantitative sensory testing (QST) can be used to assess conditioned pain modulation (CPM), allowing it to stand as a valuable clinical test for indirectly assessing the efficacy of an EPMS; it involves inducing a noxious stimulus at one location while simultaneously assessing the pain response at a distant site. Essentially, it measures the body’s ability to modulate pain perception, providing insights into the functioning of the EPMS. Individuals with a healthy EPMS typically exhibit a reduction in pain perception at the second site during the application of a conditioning stimulus, indicating effective pain inhibition. Individuals with FMS, who may likely have a dysfunction in the EPMS, have an irregular ability to modulate pain. Together with altered EPMS, individuals diagnosed with FMS display a diminished pain threshold and tolerance values compared to healthy individuals [9,10]. However, the QST metrics in FMS have high inter-subject heterogeneity [11]. Therefore, it is of major importance to understand how external factors can affect conditioned pain modulation (CPM) and other QST metrics, since it is a potential objective metric to assess EPMS and indirectly evaluate the central sensitization of chronic pain subjects [12].

Due to a broad symptomatology, individuals with fibromyalgia often receive excessive medication. Health professionals usually prescribe various pharmacological interventions such as antidepressants, opioids, and over-the-counter analgesics. Patients who are overmedicated can suffer from known drug-related adverse effects and drug interactions. Importantly, these drugs could also affect the sensory system and pain processing mechanisms in FMS patients.

The association of pain medication usage on QST measures in fibromyalgia patients is a relevant question that has not been explored in previous studies. Due to the evidence correlating antidepressant use (and other medications) with the relief of pain symptoms, researchers assumed that this relief is correlated with an improvement in sensory metrics; thus, this crucial question has not been systematically explored in the past. Moreover, poor tracking of medication use and the use of non-standardized QST measures in previous FMS studies have hampered a proper analysis. In the absence of substantial literature on the effects of pain medications on sensory outcomes in FMS patients, we designed this cross-sectional analysis aiming to explore the association of pain medication usage on QST metrics. We hypothesize that the use of pain medications, especially opioids, will be associated with a poor QST state, indexed as having lower pain threshold, higher temporal summation, and lower CPM.

## 2. Methods

### 2.1. Study Design

This research involves a cross-sectional analysis of baseline data collected as part of a randomized, double-blind clinical trial investigating the effects of transcranial direct current stimulation (tDCS) and aerobic exercise on fibromyalgia patients (NCT03371225). Baseline data, including participants’ pain medication diaries, were gathered prior to the intervention, spanning from 13 May 2019 to 26 October 2023. All participants provided informed consent, and the study received prior approval from the Institutional Review Board of Mass General Brigham’s ethics committee (2017P002524) [11]. Additional methodological details are available in the published protocol.

### 2.2. Participants

Inclusion criteria: individuals aged 18 to 65 years diagnosed with fibromyalgia pain based on the 2010 criteria of the American College of Rheumatology (ACR) [13] who have experienced pain for over 6 months, with an average intensity of at least 4 on a 0–10 Visual Analog Scale (VAS) [14], an absence of other chronic pain diagnoses, resistance to common analgesics and medications for chronic pain, and an ability to perceive sensation through stimulation. Exclusion criteria: patients with clinically significant or unstable medical or psychiatric conditions, a self-reported history of substance abuse within the past 6 months, prior substantial neurological events (e.g., traumatic brain injury, concussion) leading to neurological deficits, previous neurosurgical procedures involving craniotomy, severe depression (scoring >30 on the Beck Depression Inventory) [15], pregnancy (given the unassessed safety of tDCS in pregnant individuals (and children), although the perceived risk is considered non-significant), current use of high doses of opiates (30 mg a day of morphine or equivalent), increased risk for exercise (as defined by not meeting the criteria set by the American College of Sports Medicine (e.g., cardiovascular complication risk)), and, in this scenario, not being cleared by a licensed physician. Each participant provided written informed consent.

### 2.3. Demographic and Clinical Variables

Data on demographic and clinical characteristics were collected, including age, gender, and scores from several standardized instruments. These included the revised Fibromyalgia Impact Questionnaire (FIQR) [14] to assess health status in fibromyalgia, the Quality of Life Scale (QoL) [16] covering 15 domains relevant to chronic conditions, and the Beck Depression Inventory (BDI) [15], a 21-item measure of depression. Other assessments included visual analog scales (VASs) for anxiety, depression, and sleepiness, the Brief Pain Inventory (BPI) [17] for pain severity and interference, and the PROMIS-29 v2.0 [18] to evaluate fatigue [19], fibromyalgia, and anxiety [20]. Pain outcomes were measured using a VAS ranging from 0 (no pain) to 10 (worst pain) [21], a validated tool for assessing both acute and chronic pain.

### 2.4. Pain Medication Diary

In order to have a better overview of participants’ medication usage throughout the trial, a systematic approach was implemented to collect these data on a weekly basis. A pain medication diary was given to each patient at the beginning of the week, providing an organized tool to document their parallel treatments between visits. The diary consisted of seven sections designated for each day of the week, allowing the patients to describe how medications were administered, including the drug names, dosages, and frequencies of intake. This dynamic measure not only guaranteed a more accurate assessment of medication patterns but also allowed participants to become involved in monitoring their own treatment progress. By fostering a collaborative and transparent approach, this method contributed to a richer dataset, ultimately enhancing the trial’s efficacy in evaluating the impact of the interventions on pain management. For this cross-sectional analysis, we used the medication information for the week before the baseline visit.

### 2.5. Quantitative Sensory Testing (QST)

These variables were collected during the baseline visit of the clinical trial and were performed in sequence at the beginning of the session. This assured a temporal relationship with the pain medication information of the week before the baseline visit.

#### 2.5.1. Conditioned Pain Modulation (CPM)

For the conditioned pain modulation (CPM) protocol, heat pulses were delivered using a TSA-II Stimulator (Medoc Advanced Medical Systems, Tel-Aviv, Israel) and a 30 mm × 30 mm heat thermode (Medoc Advanced Medical Systems, Tel-Aviv, Israel) applied to the right proximal volar forearm, following protocols established by Granot et al. (2008) [22] and Nir et al. (2011) [23]. The Pain-60 temperature, defined as the temperature eliciting a pain score of 60 on a 0–100 numerical pain scale (NPS), was identified using heat stimuli at various temperatures (e.g., 43 °C, 44 °C, 45 °C) applied for 7 s each. Adjustments were made if these temperatures did not reach the Pain-60 threshold. Once established, the Pain-60 temperature was applied for 30 s, with pain ratings being recorded at 10, 20, and 30 s. Following a 5 min interval, a conditioning stimulus was applied by immersing the left hand in a water bath at 10–12 °C for 30 s while the Pain-60 stimulus was simultaneously reapplied to the forearm. Pain ratings were again recorded at 10, 20, and 30 s. The CPM response was calculated as the difference between pain ratings during the test and conditioning stimuli.

#### 2.5.2. Pain-60 and Temporal Slow Pain Summation

During the baseline assessment, we measured the sensory profile using quantitative sensory testing (QST). Our current analysis focused on Conditioned Pain Modulation (CPM), Temporal Slow Pain Summation (TSPS), and Pain-60. CPM evaluates individuals’ ability to inhibit pain, serving as an indirect estimation of the Descending Pain Modulatory System (DPMS). The technique activates a cortically regulated spinal–bulbospinal loop through the Diffuse Noxious Inhibitory Control (DNIC) [24] mechanism, resulting in the phenomenon known as “pain inhibits pain”. TSPS, following Staud et al.’s protocol [25], was utilized to rate central sensitivity using repetitive stimuli pulses [26]. Pain-60 measures the temperature evoking a pain level of 60 on a 0–100 numerical pain scale (NPS) after a series of heat stimulations. While Pain-60 serves as a reference for the CPM test, it is not directly employed in calculating the CPM score. Importantly, the scores for CPM and Pain-60 are independent of each other.

### 2.6. Statistical Analysis

Descriptive statistics were used to summarize the study sample. Continuous variables were reported as medians with interquartile ranges due to the small sample size, while categorical variables were expressed as frequencies and percentages. Analyses were performed using complete data, without imputations.

To investigate the association of pain medication usage and QST metrics (P60, TSPS, CPM), multivariable linear regression models were created using the QST metrics as dependent variables (three models). CPM and P60 were categorized using the median approach due to skewness in the distributions. TSPS was used as a continuous variable since it has a normal distribution. All dependent variables were analyzed as continuous variables. Outliers, identified as outcome or exposure measurements exceeding three standard deviations (SDs) from the mean, were excluded from the analysis if detected. The independent variables were binary variables on the use of pain medications, including opioids, antidepressants, gabapentinoids, and muscle relaxants. First, univariate models were created.

Our predefined primary outcome was CPM; the rest of the metrics were secondary; therefore, no multiple comparison adjustments were performed in order to generate hypothesis. Exposure variables were the pain medications categories consumed in the week before the baseline visit. The potential confounders were pain intensity, age, gender, and depression score; thus, they were included in our final models. Additionally, we tested the potential confounding due to medication types, but none of them survived in the models. Thus, we assumed no confounding due to medication types.

To identify the optimal multivariable regression models, univariate analyses were first conducted to identify covariates with a *p*-value < 0.2, which were then included in an initial multivariable model. A backward elimination approach was applied, sequentially removing variables with the highest non-significant *p*-values until only significant variables (*p* < 0.05), alongside forced variables, remained in the final model. Multicollinearity in the final models was assessed using variance inflation factors.

The assumptions of linearity, homoscedasticity, and normality were verified using scatterplots, residual plots, histograms, and the Shapiro–Wilk test. Regression results were summarized in tables, and all statistical analyses were conducted using R-Studio (Version 2023.06.0+421). A *p*-value < 0.05 was considered statistically significant.

## 3. Results

### 3.1. Sample Characteristics

In total, 101 fibromyalgia patients were recruited, with a median age of 50.0 years old (IQR: 39–57); there were 11 males (10.89%) and 90 females (88.12%); most completed college (82, 81.89%), while 2 (1.98%) reached middle school, 11 (10.89%) reached high school, and only 6 (5.94%) have a doctorate degree. The majority of the participants, 76, declared themselves as white (75.25%), 1 (0.99%) declared themselves to be American Indian or Alaska Native, 3 (2.97%) as Asian, 9 (8.91%) as Black or African American, 7 (6.93%) declared having more than one race, and 5 (4.95%) did not report their race. Fibromyalgia duration had a median of 10 months (IQR: 4–17), and the Fibromyalgia Impact Questionnaire had a mean of 54.51 (SD: 18.10). Regarding medication usage, 11 patients were on opioids, 50 were on antidepressants (43 using duloxetine, 5 milnacipran, and 2 amitriptyline), 25 were on gabapentinoids, and 26 were on muscle relaxants. Table 1 summarizes the sample characteristics, and a similar description of the sample is reported in a previous analysis from our group [27].

### 3.2. Dependent Variables Characteristics

Table 2 presents the characteristics of our three dependent variables: CPM mean = 1.31 (95% CI: 0.97–1.64) and SD = 1.72; TSPS = −0.21 (95% CI: −0.52–0.09) and SD = 1.57; P60 = median = 47 and IQR = 46–48). The dependent variables’ distribution can be found in Figure 1.

### 3.3. Univariate Analysis

Initially, we conducted a univariate analysis to identify medications potentially linked to the observed outcomes, setting the *p*-value threshold at 0.25. No outlier was identified and all observations were included. In this analysis, for the outcome P60, only gabapentinoids fell within the threshold with a *p*-value of 0.18. For TSPS, muscle relaxants showed significance, with a *p*-value of 0.04, along with corticosteroids at a *p*-value of 0.10. In the case of CPM, antidepressants and gabapentinoids exhibited *p*-values of 0.13 and 0.15, respectively. Interestingly, the use of opioids did not present a statistically significant correlation with any of the outcomes studied, implying they may not influence the results of our research. Subsequently, we implemented a multivariable regression analysis to adjust for any confounding variables.

### 3.4. Multivariable Analysis

Table 3 presents the models adjusted for pain intensity, age, gender, and depression score.

### 3.5. Conditioned Pain Modulation (CPM) Models

CPM was categorized into binary categories (either below or above 20% of improvement). Then, we found a negative relation between CPM and the use of antidepressants OR = 0.39 (95% CI = 0.17–0.91), indicating that, the higher the use of antidepressants, the lower the CPM. This finding is adjusted for pain intensity, age, gender, and depression score.

### 3.6. Temporal Slow Pain Summation (TSPS) Models

We found a negative relation between TSPS and the use of muscle relaxants β coefficient = 0.72 (95% CI = 0.0021–1.4431), indicating that, the higher the use of muscle relaxants, the lower the TSPS. This finding is adjusted for pain intensity, age, gender, and depression score.

### 3.7. Pain-60 (P60) Models

P60 was categorized into binary categories (either below 48 °C or equal to 48 °C). Then, we found a positive relation between P60 and the use of gabapentinoids, OR = 2.83 (95% CI = 1.03–8.24), indicating that, the higher the use of gabapentinoids, the higher the p60. This finding is adjusted for pain intensity, age, gender, and depression score.

## 4. Discussion

The present study, with a population of 101 fibromyalgia patients using daily pain medications, including antidepressants, muscle relaxants, gabapentinoids, and opioids, aims to understand how different usage of drug classes are associated QST parameters in FMS patients. Fibromyalgia is commonly associated with dysfunction in the descending pain inhibitory pathways of the Endogenous Pain Modulatory System (EPMS) [11]. The literature shows that antidepressants are the most utilized medications among fibromyalgia patients [28,29]. Evidence-based guidelines recommend the use of antidepressants for fibromyalgia management [30,31]. Contrary to what was expected, our results correlate antidepressant usage with worsened CPM, indicating that the utilization of antidepressants may signify an increased susceptibility to pain rather than its suppression. Our study showed a potential negative impact on their already compromised Endogenous Pain Modulatory System (EPMS). Similarly, this analysis also presents a negative association between muscle relaxant usage and Temporal Slow Pain Summation (TSPS), suggesting a worsening effect in pain sensitization. On the other hand, a positive association is observed between the usage of gabapentinoids and the Pain-60 test temperature (P60), indicating that higher gabapentinoid usage is associated with higher pain thresholds. This discovery is particularly relevant since those are the main drug classes prescribed for this syndrome in clinical practice [32].

### 4.1. The Use of Antidepressants

Conditioned Pain Modulation (CPM) was found to have a negative correlation with the use of antidepressants in the present study. CPM was categorized into binary categories based on a 20% improvement threshold. It is believed that the acting of the Descending Pain Inhibitory System (DPIS) can be increased by drugs that have a role in the monoaminergic system, such as antidepressants with dual or tricycle effects [33]. Noradrenaline’s crucial role in inhibiting chronic pain was emphasized by recent studies on animal models. Initially, the direct inhibition of chronic pain through α2-adrenergic receptors is achieved by increasing noradrenaline in the spinal cord through reuptake inhibition. Following this, the influence of raising noradrenaline on the locus coeruleus improves the function of a compromised descending noradrenergic inhibitory system. Serotonin and dopamine may amplify the noradrenergic effects, contributing to the inhibition of chronic pain [34]. Contrary to what was previously expected, this analysis noticed that higher usage of antidepressants is associated with reduced efficacy of CPM. This finding was maintained after adjusting for variables such as pain intensity, age, gender, and depression. It challenges our previous expectations that antidepressants enhance CPM, as it has been found that they potentially contribute to its decrease. The implications of this discovery illustrate the need for broader research into the mechanisms through which antidepressants influence sensory outcomes.

### 4.2. Skeletal Muscle Relaxants

We found a negative relationship between Temporal Slow Pain Summation (TSPS) and the use of muscle relaxants, suggesting that increased usage of muscle relaxants is linked to a reduction in TSPS.

This inverse correlation prompts a re-evaluation of the role of muscle relaxants in modulating pain processing in fibromyalgia. Muscle relaxants for skeletal muscles are frequently prescribed medications designed to alleviate muscle spasms linked to acute and painful musculoskeletal conditions, such as fibromyalgia. Skeletal muscle relaxants employ mechanisms such as GABAergic modulation, centrally acting alpha-2 agonism [35], and nicotinic receptor blockade to reduce muscle tone. Dantrolene inhibits calcium release [36], while some, like cyclobenzaprine, antagonize NMDA receptors. The peripheral neuromuscular blockade, as seen with baclofen, involves GABA-B receptor activation, reducing spinal reflex hyperexcitability, and alleviating muscle spasms [37]. Each mechanism targets specific pathways in the central nervous system or muscle cells, collectively contributing to the therapeutic effects of skeletal muscle relaxants. Despite their common prescription, the utilization of these medications in treating musculoskeletal conditions is a subject of controversy due to a limited understanding of their mechanisms of action, insufficient clinical evidence supporting their efficacy, and the high incidence of side effects within the entire drug class [38]. The American Pain Society and the American College of Physicians advocate the initial use of acetaminophen and nonsteroidal anti-inflammatory drugs (NSAIDs) for managing acute low back pain while suggesting skeletal muscle relaxants as an alternative. This guidance is grounded in the existing literature, which indicates that skeletal muscle relaxants do not outperform NSAIDs in effectiveness for patients with acute back pain [39]. Our findings are significant as they reiterate that the use of muscle relaxants for fibromyalgia may not be the optimal recommendation. In our study, their use is linked to worse sensory outcomes, indicating that individuals who take this drug class in our group experience more pain. This may contribute to the refinement of treatment strategies for fibromyalgia patients.

### 4.3. Gabapentinoids Usage

Our study presents a positive association between Pain-60 Test Temperature (P60) and the use of gabapentinoids, presenting a notable finding of potential clinical significance. Pregabalin and gabapentin, known as gabapentinoids, primarily function as anticonvulsant medications. In the last ten years, there has been a growing trend in their prescription for pain management. While usually recommended for treating neuropathic pain in adults, they are frequently employed off-label to address various chronic pain conditions, including fibromyalgia [40]. According to a Cochrane review, high-quality evidence indicates that 10% of individuals with moderate to severe fibromyalgia who are administered pregabalin at a daily dosage of 300–600 mg achieve a notable 30–50% alleviation in pain within the span of 12 to 26 weeks [41]. Although a recent meta-analysis published on pain shows that there is still a lack of evidence regarding the long-term effectiveness of gabapentin in fibromyalgia patients, and considering the relevant side effects linked to alpha-2-delta-ligands, it is crucial to reassess their impact at regular intervals to determine whether the benefits outweigh the side effects [42]. Gabapentinoids modulate neurotransmitter release by binding to the α2δ subunit of voltage-gated calcium channels. This interaction inhibits the entry of calcium ions into presynaptic neurons, leading to a reduced release of excitatory neurotransmitters like glutamate. Gabapentin’s ability to selectively target hyperexcitable neurons contributes to its anticonvulsant effects and effectiveness in managing neuropathic pain. While it indirectly influences the GABAergic system, its primary mechanism involves calcium channel modulation, making it valuable in conditions characterized by abnormal neuronal excitability [43]. Our study reveals that higher utilization of gabapentinoids is linked to an increased P60. This correlation agrees with the conventional perspectives on the influence of gabapentinoids on pain perception, suggesting a potential role in diminishing pain responses in fibromyalgia patients. A potential explanation is an enhanced TRPV1-dependent signaling that is important in thermal nociception, as a previous study has shown that gabapentin could suppress hyperalgesia after heat-capsaicin sensitization [44]. Consistently, in our study we found that gabapentin increased the pain sensitivity to a heat stimulus. The need for further research into these mechanisms driving this relationship is evident, as it could deepen our understanding of how gabapentinoids impact pain processing. These findings contribute to the broader discourse on pharmacological interventions for sensory outcomes in fibromyalgia, highlighting the potential of gabapentinoids as a targeted therapeutic approach for managing pain response.

### 4.4. The Role of Opioids Dose

It is known that long-term opioid therapy can increase dysfunction in the pain inhibitory system (DPIS). A survey conducted recently on the national level suggests that approximately 3–4% of adults in the United States are undergoing long-term opioid therapy. However, opioid treatments lead to various side effects, including analgesic tolerance and opioid-induced hyperalgesia (OIH). These occurrences are closely connected to the dysfunction of the descending pain inhibitory system (DPIS) [45]. Besides dependence, opioids can trigger an unforeseen rise in pain sensitivity. This occurrence is identified as hyperalgesia induced by opioids (OIH). The biological mechanisms underlying pronociceptive activity, including activation of adenylate cyclase, N-methyl-D-aspartate (NMDA)-type glutamate receptor activation, and the release of pronociceptive peptides such as dynorphin A and neuropeptide FF, are presumed to be responsible for OIH. Célèrier et al. [46] describe OIH in terms of the “opponent process theory”, where the exogenous central effect of the drug (antinociceptive activity) is counterbalanced by an endogenous response (pronociceptive activity) [47]. The exact mechanism of this paradoxical response is still unclear but is likely multifactorial [48].

In our sample, the use of opioids below this dosage was not linked to an altered QST outcome. A limitation of our study is an exclusion criterion of usage of >30 mg of morphine equivalents per day in this population. Consequently, we cannot exclude that a high dosage of opioid usage would likely alter sensory outcomes in fibromyalgia patients. Additional research is needed, with a specific focus on higher opioid dosages, for a better understanding of their potential impact on sensory outcomes within fibromyalgia patients.

Our results explain the relationship between common pain medications used by fibromyalgia patients and their effects on the quantitative sensory testing (QST) outcomes. It brings attention to the important interference of pharmacological interventions on pain perception in individuals who have a previously compromised Endogenous Pain Modulatory System and must deal with high levels of pain daily. The main insights of this study focus on the relationship between antidepressants and QST outcomes, contributing to a broader knowledge regarding medication use and neurophysiological function in fibromyalgia. We showed a clear association between the usage of antidepressants, muscle relaxants, and gabapentin with QST profiles in fibromyalgia patients—exposing an interference in assessment outcomes and sensory responses within this population. The use of antidepressants was connected to a worsened conditioned pain modulation (CPM) function. This result persisted after adjustments for pain and depression levels, suggesting an impact of antidepressants on the central pain modulation pathways in fibromyalgia. The usage of opioids below that range did not show a statistically significant link to altered QST profiles in our sample. This observation suggests that, within this dosage, opioids may not significantly impact the neurophysiological mechanisms assessed by QST in fibromyalgia patients.

### 4.5. Implications on Physiotherapy Practice

Many fibromyalgia patients are prescribed multiple medications with limited objective documentation of their benefit, often without systematic attempts at therapeutic optimization [49]. This pattern of prescribing may contribute to suboptimal treatment outcomes, potential drug-related adverse effects, and unintended influences on sensory processing, as evidenced by our findings. The observed associations between medication use and altered quantitative sensory testing (QST) outcomes highlight the need for a more individualized approach to fibromyalgia treatment. Rather than relying on a trial-and-error strategy with broad pharmacological interventions, clinicians should consider tailoring therapy based on objective assessments of pain modulation and patient-specific responses. These findings reinforce the importance of precision medicine in fibromyalgia management, advocating for a data-driven approach to optimize therapeutic strategies while minimizing unnecessary polypharmacy.

Fibromyalgia treatment still relies on pharmacologic measures to manage the symptoms of pain. Although the proven benefits, they can cause various side effects and are not recommended independently [50]. The optimized treatment strategy is a multidisciplinary approach involving both medication and physical therapy, enhancing the overall quality of life by reducing pain, improving physical function, and promoting mental well-being [51]—physical exercise is essential in managing fibromyalgia to improve fatigue and pain. Beyond those mentioned, other interventions, such as non-invasive brain stimulation techniques, are also being used. As mentioned before, medications, mainly antidepressants and gabapentin, still play an essential role in the treatment plan for these patients [42,52]. In this study, we found a result correlating the usage of antidepressants with a worsened CPM, suggesting that this medication class could potentially be affecting the body’s endogenous pain modulation system. By addressing symptoms non-pharmacologically, patients may experience a reduced need for medications, which can minimize side effects and potential long-term impacts on the pain modulation system. Regarding NIBS, transcranial direct current stimulation (tDCS) is one of the prominent techniques, and the evidence shows moderate to large effects on chronic pain and should be considered [53]. Based on these findings, we may infer that incorporating physiotherapy into fibromyalgia management strategies to reduce the use of medications demonstrates its potential to alleviate symptoms and improve functional capacity.

### 4.6. Limitations

This study has some limitations that should be considered when interpreting the findings. First, the cross-sectional design precludes causal inferences regarding the relationship between medication use and quantitative sensory testing (QST) outcomes. It is unclear whether the observed sensory alterations are a consequence of medication use, a reflection of underlying fibromyalgia pathophysiology, or both. Second, the study relied on self-reported medication use, which may introduce recall bias and does not account for adherence, dosage variations, or duration of treatment—all of which could influence sensory outcomes. Additionally, we did not assess the impact of medication dosage, which may have differential effects on sensory modulation. Third, our study was exploratory in nature and did not include multiple comparison adjustments, increasing the risk of Type I errors. Therefore, our findings should be interpreted as hypothesis-generating, requiring validation in future studies. Moreover, while we categorized medication use into broad classes, we did not include specific subtypes of antidepressants, opioids, or other analgesic medications, which may have distinct effects on pain modulation. Finally, while we adjusted for potential confounders, unmeasured variables such as psychological factors (e.g., depression, anxiety), lifestyle behaviors, or comorbid conditions may have influenced our results. The study also did not assess combination therapy effects, which are common in fibromyalgia management and could have additive or interactive effects on pain modulation. Future longitudinal studies with larger, well-characterized cohorts and controlled medication trials are needed to clarify the causal relationships between pain medications and sensory processing in fibromyalgia.

## 5. Conclusions

The findings generated in this study showcase the critical need for personalized pharmacological approaches in fibromyalgia treatment. The associations observed with antidepressants and QST parameters point out the importance of considering refinement when selecting and prescribing this drug class to fibromyalgia patients. Since the directionality of the association is not clear with our study, future longitudinal research focused on the mentioned interference of antidepressants on sensory outcomes within those patients is needed to improve and refine treatment strategies for individuals living with FMS.

## Figures and Tables

**Figure 1 neurosci-06-00015-f001:**
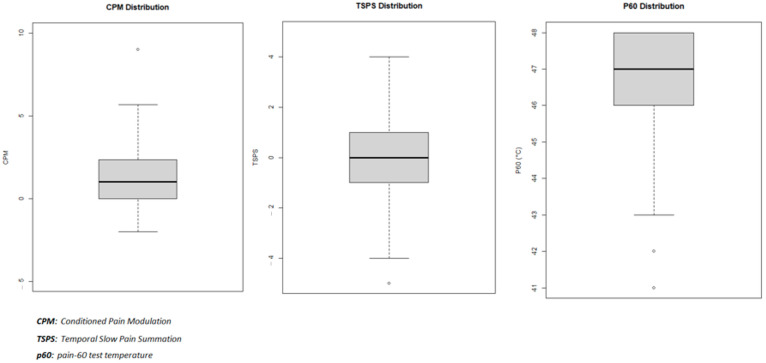
Description of quantitative sensory testing variables.

**Table 1 neurosci-06-00015-t001:** Sample Demographic Characteristics.

Age Years: Median, IQR	50.0	39.0–57.0
Gender: number, %	Males: 11	10.89%
	Females: 89	88.12%
Education: number, %	Middle School = 2	1.98%
	High School = 11	10.89%
	College = 82	81.89%
	Doctorate degree = 6	5.94%
Ethnicity: number, %	American Indian or Alaska Native: 1	0.99%
	Asian: 3	2.97%
	Black or African American: 9	8.91%
	Native Hawaiian or Other Pacific Islander: 0	0%
	White: 76	75.25%
	More than one race: 7	6.93%
	Unknown or not reported: 5	4.95%
Fibromyalgia Duration (months): median, IQR	10	4–17
Fibromyalgia Impact Questionnaire: Mean, SD	54.51	18.10
Body Mass Index: Median, IQR	28.32	22.86–32.92

Abbreviations: SD: standard deviation; IQR: interquartile range.

**Table 2 neurosci-06-00015-t002:** QST variables description.

Variable	Mean or Median (95% CI)	SD or IQR
Conditioned Pain Modulation: Median, IQR	1.0 (0.7–1.5)	0.0–2.3
Temporal Summation of Pain: Median, IQR	0.0 (0.0–0.0)	(−1)–(+1)
Pain-60: Median, IQR	47 (47–48)	46–48

Abbreviations: CI: confidence interval; SD: standard deviation; IQR: interquartile range.

**Table 3 neurosci-06-00015-t003:** Multivariable Analysis.

Conditioned Pain Modulation	OR (95% CI)	*p*-Value	Pseudo R^2^
Antidepressants	0.39 (0.17–0.91)	0.03	0.06
Temporal Slow Pain Summation	β coefficient (95% CI)	*p*-Value	Adjusted R^2^
Muscle Relaxants	0.72 (0.0021–1.4431)	0.049	0.03
Pain-60	OR (95% CI)	*p*-Value	Pseudo R^2^
Gabapentinoids	2.83 (1.03–8.24)	0.048	0.09

Abbreviations: OR (Odds Ratio), CI (confidence interval).

## Data Availability

Data will be available after completion of the ongoing clinical trial following funding agency (NIH) procedures.

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
