# Peer review of "The Association of Pain Medication Usage and Quantitative Sensory Testing Outcomes in Fibromyalgia Patients: A Secondary Data Analysis"

_neurosci, 2025, doi:10.3390/neurosci6010015_

Round 1
Reviewer 1 Report
Comments and Suggestions for Authors
Thank you for the opportunity to review this interesting work. This is a difficult condition to manage. Many patients presenting to specialist services have already been tried on multiple medications, and have their own beliefs about them, with limited formal documentation of benefit. It requires new thinking.
The introduction sets out the background clearly with appropriate supporting literature.
The methods are clearly explained and presented and are reproducible, although the statement in line 195 'Outliers were defined as values greater than 3 standard deviations (SD) away from the mean scores and removed from the analysis if detected' would benefit from more clarification.
The results are well set out.
The discussion, divided by medications classes is a useful approach, but noes not appear to link directly to the conclusion. I feel it would be helpful to briefly develop that many of these patients have been given multiple medications with poor objective documentation of benefit or any attempt at therapeutic optimization, this would reinforce the message of individualization of therapy.
Comments on the Quality of English LanguageThe quality of English is excellent, requiring only very minor editorial changes for an international audience.
Author Response
Response letter
Reviewer 1:
Thank you for the opportunity to review this interesting work. This is a difficult condition to manage. Many patients presenting to specialist services have already been tried on multiple medications and have their own beliefs about them, with limited formal documentation of benefit. It requires new thinking.
Response: Thank you for your feedback.
The introduction sets out the background clearly with appropriate supporting literature.
Response: Thank you. We appreciate your feedback.
The methods are clearly explained and presented and are reproducible, although the statement in line 195 'Outliers were defined as values greater than 3 standard deviations (SD) away from the mean scores and removed from the analysis if detected' would benefit from more clarification.
Response: Thank you. We have clarified that we predefined this definition for outlier values, but we did not identify important outliers; therefore, we did not remove any value from our analysis.
The results are well set out.
Response: Thank you.
The discussion, divided by medications classes is a useful approach, but noes not appear to link directly to the conclusion. I feel it would be helpful to briefly develop that many of these patients have been given multiple medications with poor objective documentation of benefit or any attempt at therapeutic optimization, this would reinforce the message of individualization of therapy.
Response: Thank you for your comments. We have added a new paragraph in the discussion that addresses your commentary that multiple medications are used for fibromyalgia patients but with variable effects, suggesting a need for therapeutic optimization, especially the implementation of non-pharmacological interventions in fibromyalgia care.
Reviewer 2 Report
Comments and Suggestions for Authors
The authors report surprising new data showing an association between medication usage and quantitative sensory testing outcomes in Fibromyalgia (FM) patients.
I have only few minor comments that may improve the manuscript.
At what time of the day CPM protocol was performed? Is there prior evidence for diurnal variation in CPM testing outcomes?
Which specific antidepressant were used by FM patients in the study? Duloxetine, milnacipran, amitriptyline, others? Did all antidepressants worsen the CPM similarly?
Apparently heat pain/heat sensitivity is a significant burden in FM. Perhaps the authors could cite earlier healthy volunteer study in which gabapentin was shown to attenuate heat/capsaicin-induced cutaneous hyperalgesia (PMID: 12131110) whereas thermal nociception in normal skin was unchanged. It is now well established that the TRPV1 ion channel is a heat/capsaicin receptor. Could the finding that gabapentin use was associated with elevated Pain-60 suggest that TRPV1-dependent signaling is enhanced in FM?
Author Response
Reviewer 2:
The authors report surprising new data showing an association between medication usage and quantitative sensory testing outcomes in Fibromyalgia (FM) patients. I have only a few minor comments that may improve the manuscript.
Response: Thank you for your comments and feedback.
At what time of the day CPM protocol was performed? Is there prior evidence for diurnal variation in CPM testing outcomes?
Response: No previous evidence suggests a diurnal variation in CPM metrics because this compensatory mechanism is fundamental for pain adaptation and is not expected to be highly influenced by the sleep-wake cycle (see reference below). Our protocol was primarily performed between 9 am to 5 pm, which theoretically would not be influenced by circadian shifts or cortisol peaks, this ensures the quality of our measurements.
Reference: Matre D, Andersen MR, Knardahl S, Nilsen KB. Conditioned pain modulation is not decreased after partial sleep restriction. European Journal of Pain. 2016 Mar;20(3):408-16.
Which specific antidepressant were used by FM patients in the study? Duloxetine, milnacipran, amitriptyline, others? Did all antidepressants worsen the CPM similarly?
Response: Thank you for your comment. Our sample used mainly duloxetine (43 subjects out of 50 subjects using antidepressants). Additionally, five used milnacipran, and two used amitriptyline. Therefore, our results can only be interpreted in the context of using these three medications. However, because those are the ones FDA-approved for Fibromyalgia, we generalized the findings to antidepressants commonly prescribed to treat this condition. We agree with your point that future studies should explore the role of other antidepressant classes. We have added these points to our results and limitations section.
Apparently heat pain/heat sensitivity is a significant burden in FM. Perhaps the authors could cite earlier healthy volunteer study in which gabapentin was shown to attenuate heat/capsaicin-induced cutaneous hyperalgesia (PMID: 12131110) whereas thermal nociception in normal skin was unchanged. It is now well established that the TRPV1 ion channel is a heat/capsaicin receptor. Could the finding that gabapentin use was associated with elevated Pain-60 suggest that TRPV1-dependent signaling is enhanced in FM?
Response: Thank you for your suggestion. We agree that the molecular mechanism behind our findings of gabapentin's effect on heat-based QST variables could be attributed to enhanced TRPV1-dependent signaling. We have added this hypothesis to our discussion section on gabapentin effects.
Reviewer 3 Report
Comments and Suggestions for Authors
This is a secondary analysis of baseline data of an RCT, aiming to explore the association between medication usage and sensory alteration in patients with fibromyalgia. My comments are as follows:
Major comments:
1. Please indicate the setting of the RCT in the abstract, multi- or single- center? Referral hospital or primary care? Which city is it?
2. I’m confused about the timeline. When did the QST conduct? In “Medication diary”, you mentioned data were collected weekly, so what time did your cross-sectional analysis focus on? If you aimed to analyze the association between medication and QST, the QST should be assessed after the administration of medication, so did you re-assess QST in patients who changed their regimen? Please provide this details.
3. Please clarify the definition of primary outcome, exposure, and confounders in Methods. Since you analyzed data of multiple QST tests, if all of them were your primary outcomes, you needed to adjust for the statistical power of multiple comparisons.
3. If you did not analyze the association between the dose of medications and QSTs, this should be a limitation. The effects of medications on sensation may differ across dosages.
4. To analyze the association, multivariable models should be employed to adjust for confounding effects. When selecting variables for inclusion in the multivariable model, it is essential to base the selection on confounder criteria. A key criterion is the imbalance in the distribution of confounders across exposure groups. I assume that each medicine can be a confounder of other medicines. Therefore, I recommend assessing this by grouping patients based on one medication and calculating the standardized difference of the other medicine as well as clinical characteristics, which should be presented as Table 1. Variables with a standardized difference greater than 0.1 should be included in the multivariable model, rather than focusing solely on variables significantly associated with the outcome.
5. In table 2, what are the “clinical characteristics”?
6. Please state the limitations of this study in Discussion as a paragraph.
Minor concerns:
1. Please change “multivariate” to “multivariable” throughout the manuscript.
2. Please avoid using the abbreviation “QST” in the title.
3. Please indicate this is a secondary analysis of RCT, instead of saying “cross-sectional analysis” in the title.
4. The front size is smaller in the 3rd paragraph of the introduction section.
5. “Outliers were defined as values greater than 3 standard deviations (SD) away from the mean scores and removed from the analysis if detected.” Did you exclude the outlier measurement (still keep this patient) or exclude patients with any outliers? Please clarify.
6. Please provide the title and figure legend of Fig1.
7. Please explain the abbreviations in the footnotes of the tables.
Author Response
Reviewer 3:
This is a secondary analysis of baseline data of an RCT, aiming to explore the association between medication usage and sensory alteration in patients with fibromyalgia. My comments are as follows:
Major comments:
- Please indicate the setting of the RCT in the abstract, multi- or single- center? Referral hospital or primary care? Which city is it?
Response: We added those details into the abstract.
- I’m confused about the timeline. When did the QST conduct? In “Medication diary”, you mentioned data were collected weekly, so what time did your cross-sectional analysis focus on? If you aimed to analyze the association between medication and QST, the QST should be assessed after the administration of medication, so did you re-assess QST in patients who changed their regimen? Please provide this details.
Response: Thank you for raising this point. Our cross-sectional analysis was focused on the baseline visit. In this visit the QST metrics were performed at the beginning of the session. Also, that day patients handled the pain medication diary of the week before the baseline visit. For our analysis, we used the CPM test of the baseline visit after a week of medication use (reported in the medication diary), which was very stable across the participants because it was a inclusion criteria for our clinical trial. Therefore, we analyzed the association having a temporal sequence and minimizing the change in medication regimen. We provided more details in the methods section.
- Please clarify the definition of primary outcome, exposure, and confounders in Methods. Since you analyzed data of multiple QST tests, if all of them were your primary outcomes, you needed to adjust for the statistical power of multiple comparisons.
Response: Thank you for your comment. We clarified the outcomes in the methods section. Since we have only one primary outcome (CPM), we decided not to adjust for multiple comparison due to the exploratory nature of our study. We added this point to our limitations section.
- If you did not analyze the association between the dose of medications and QSTs, this should be a limitation. The effects of medications on sensation may differ across dosages.
Response: Thank you we added this point to our limitations section.
- To analyze the association, multivariable models should be employed to adjust for confounding effects. When selecting variables for inclusion in the multivariable model, it is essential to base the selection on confounder criteria. A key criterion is the imbalance in the distribution of confounders across exposure groups. I assume that each medicine can be a confounder of other medicines. Therefore, I recommend assessing this by grouping patients based on one medication and calculating the standardized difference of the other medicine as well as clinical characteristics, which should be presented as Table 1. Variables with a standardized difference greater than 0.1 should be included in the multivariable model, rather than focusing solely on variables significantly associated with the outcome.
Response: Thank you for your suggestions. As you mentioned, we performed multivariable regression models using confounder criteria based on previous literature and the change in estimate criteria following a purposeful selection of variables (references below). We defined that pain intensity, age, gender, and depression score were potential confounders. Therefore, our models were adjusted for these variables. Regarding the type of medications, we decided not to use a grouping strategy because of the detrimental effects on statistical power (creating a subset of our database and increasing imprecision). Thus, the multivariable models included the medication variables as potential confounders. However, when selecting informative variables (cutoff of change in estimate of 0.1), none of the medication categories survived. So, our data did not suggest a potential confounding or multicollinearity problem between the medication types. We added a note in methods to clarify that we tested for this plausible confounding effect.
References:
Bursac Z, Gauss CH, Williams DK, Hosmer DW. Purposeful selection of variables in logistic regression. Source code for biology and medicine. 2008 Dec;3:1-8.
Lee PH. Should we adjust for a confounder if empirical and theoretical criteria yield contradictory results? A simulation study. Scientific reports. 2014 Aug 15;4(1):6085.
- In table 2, what are the “clinical characteristics”?
Response: We apologize for the typo, the title of table 2 is “QST variables description.” We have corrected the text.
- Please state the limitations of this study in Discussion as a paragraph.
Response: Thank your for your suggestion. We have added a limitations section.
Minor concerns:
- Please change “multivariate” to “multivariable” throughout the manuscript.
Response: Corrected.
- Please avoid using the abbreviation “QST” in the title.
Response: Corrected.
- Please indicate this is a secondary analysis of RCT, instead of saying “cross-sectional analysis” in the title.
Response: Corrected.
- The front size is smaller in the 3rd paragraph of the introduction section
Response: Corrected.
- “Outliers were defined as values greater than 3 standard deviations (SD) away from the mean scores and removed from the analysis if detected.” Did you exclude the outlier measurement (still keep this patient) or exclude patients with any outliers? Please clarify.
Response: Thank you for your question, we defined the identification of outlier measurements and keeping the patient with the available measurements. However, we did not identify outliers in our outcome or exposure metrics, so we did not exclude any data point in our analysis. We have clarified this in the result and methods sections.
- Please provide the title and figure legend of Fig1.
Response: Corrected.
- Please explain the abbreviations in the footnotes of the tables.
Response: Corrected.
Round 2
Reviewer 3 Report
Comments and Suggestions for Authors
The authors successfully addressed my concerns. I appreciate their efforts.